# Genome-Wide DNA Methylation Profiling Solves Uncertainty in Classifying *NSD1* Variants

**DOI:** 10.3390/genes13112163

**Published:** 2022-11-19

**Authors:** Marco Ferilli, Andrea Ciolfi, Lucia Pedace, Marcello Niceta, Francesca Clementina Radio, Simone Pizzi, Evelina Miele, Camilla Cappelletti, Cecilia Mancini, Tiziana Galluccio, Marco Andreani, Maria Iascone, Luigi Chiriatti, Antonio Novelli, Alessia Micalizzi, Marta Matraxia, Lucia Menale, Flavio Faletra, Paolo Prontera, Alba Pilotta, Maria Francesca Bedeschi, Rossella Capolino, Anwar Baban, Marco Seri, Corrado Mammì, Giuseppe Zampino, Maria Cristina Digilio, Bruno Dallapiccola, Manuela Priolo, Marco Tartaglia

**Affiliations:** 1Area di Ricerca Genetica e Malattie Rare, Ospedale Pediatrico Bambino Gesù, IRCCS, 00146 Rome, Italy; 2Department of Pediatric Onco-Haematology and Cell and Gene Therapy, Ospedale Pediatrico Bambino Gesù, IRCCS, 00146 Rome, Italy; 3Medical Genetics Laboratory, ASST Papa Giovanni XXIII, 24127 Bergamo, Italy; 4Unità di Genetica Medica, Grande Ospedale Metropolitano “Bianchi-Melacrino-Morelli”, 89124 Reggio Calabria, Italy; 5Translational Cytogenomics Research Unit, Ospedale Pediatrico Bambino Gesù, IRCCS, 00146 Rome, Italy; 6Medical Genetics Unit, IRCCS Burlo Garofolo, 34137 Trieste, Italy; 7Maternal-Infantile Department, University Hospital of Perugia, 06156 Perugia, Italy; 8Auxo-Endocrinology, Diabetology and Medical Genetic Unit, Department of Paediatrics, ASST Spedali Civili, 25123 Brescia, Italy; 9Medical Genetic Unit, Fondazione IRCCS Cà Granda Ospedale Maggiore Policlinico Clinica Mangiagalli, 20122 Milan, Italy; 10Medical and Surgical Department of Pediatric Cardiology, Ospedale Pediatrico Bambino Gesù, IRCCS, 00146 Rome, Italy; 11U.O. Genetica Medica, IRCCS Azienda Ospedaliero-Universitaria di Bologna, 40138 Bologna, Italy; 12Dipartimento Scienze della Vita, Università Cattolica del Sacro Cuore, 00168 Rome, Italy

**Keywords:** DNA methylation, overgrowth, *NSD1*, Sotos syndrome, genomic variant classification, VoUS validation, differential diagnosis

## Abstract

Background: Inactivating *NSD1* mutations causing Sotos syndrome have been previously associated with a specific genome-wide DNA methylation (DNAm) pattern. Sotos syndrome is characterized by phenotypic overlap with other overgrowth syndromes, and a definite diagnosis might not be easily reached due to the high prevalence of variants of unknown significance (VoUS) that are identified in patients with a suggestive phenotype. Objective: we performed microarray DNAm profiling in a set of 11 individuals with a clinical suspicion of Sotos syndrome and carrying an *NSD1* VoUS or previously unreported variants to solve uncertainty in defining pathogenicity of the observed variants. The impact of the training cohort size on sensitivity and prediction confidence of the classifier was assessed. Results: The Sotos syndrome-specific DNAm signature was validated in six individuals with a clinical diagnosis of Sotos syndrome and carrying *bona fide* pathogenic *NSD1* variants. Applying this approach to the remaining 11 individuals with *NSD1* variants, we succeeded in confirming pathogenicity in eight subjects and excluding the diagnosis of Sotos syndrome in three. The sensitivity and prediction confidence of the classifier based on the different sizes of the training sets did not show substantial differences, though the overall performance was improved by using a data balancing strategy. Conclusions: The present approach solved uncertainty in cases with *NDS1* VoUS, further demonstrating the clinical utility of DNAm profiling.

## 1. Introduction

Overgrowth syndromes are characterized by a global or regional excess of growth compared to equivalent body parts or age-related peer groups [1]. Sotos syndrome (OMIM #117550) is an autosomal dominant condition characterized by a triad including overgrowth (increased height, macrosomia, and macrocephaly), distinctive facial features, and developmental delay (DD)/intellectual disabilities (ID) [2]. The typical facies include dolichocephaly, broad and prominent forehead, sparse fronto-temporal hair, downslanted palpebral fissures, long and narrow face, and long/pointed chin. Sotos syndrome is caused by heterozygous inactivating variants in the nuclear receptor-binding SET domain protein gene (*NSD1*, OMIM #606681), which encodes a histone methyltransferase that preferentially methylates lysine 36 of histone H3 and lysine 20 of histone H4, either negatively or positively influencing transcription, depending on the cellular context [3,4]. *NSD1* haploinsufficiency is supported by the finding of microdeletions of the 5q35 chromosomal region in a subset of patients [3,5]. The overall prevalence of Sotos syndrome is estimated at 1 in 14,000 live births. More than 95% of the cases arise from de novo mutations; however, few cases of familial segregation have been reported [6,7]. The majority of *NSD1* pathogenic variants are point mutations causing an early termination of transcription (nonsense or frameshift variants), though changes causing amino acid substitutions within specific domains of the protein are also observed [4].

*NSD1* is characterized by low tolerance to LoF variants (pLI = 1). On the other hand, missense variants seem to be constrained to a lower extent (Z = 3.41), with pathogenic changes showing a clear positional clustering [8,9]. Among the 519 missense/splice site variants reported in gnomAD (https://gnomad.broadinstitute.org; accessed on 5 October 2022), 396 are classified as variant of unknown significance (VoUS) or have a conflicting interpretation in ClinVar (https://www.ncbi.nlm.nih.gov/clinvar/ (accessed on 5 October 2022)). The pathogenic missense variants largely map within functional domains implicated in chromatin regulation at the 3′ of the gene (i.e., PHD, SAC, and SET domains), and predominantly occur at the consensus cysteine and histidine residues that define these domains, with few exceptions [4,9]. The uncertainty of an increasing number of *NSD1* variants that are routinely revealed by massive parallel sequencing in the diagnostic setting is generally solved by segregation analysis, or might require dedicated functional validation, which is further more relevant in patients with atypical or incomplete presentation of the disorder. 

A large proportion of overgrowth syndromes is caused by pathogenic variants in genes that encode proteins involved in epigenetic regulation, indicating that the functional disruption of the epigenetic machinery is a recurrent mechanism causing these disorders [4,10,11,12]. Epigenetic marks, including DNA methylation (DNAm) and histone modifications, have emerged as crucial genome-wide regulatory events modulating the transcriptome temporally and spatially to drive normal developmental and cellular processes [13]. Recently, dozens of genes involved, directly or indirectly, in chromatin remodeling have been associated, when mutated, with a unique genome-wide DNAm profile (known as “episignature”), which is defined as the cumulative DNAm pattern occurring at multiple CpG dinucleotides across the genome [14,15,16,17]. These disease-specific DNAm signatures are detectable in peripheral blood despite the variable nature and complexity of diseases and the high variance characterizing the DNAm of genomes from different cells and tissues [18]. For this reason, DNAm testing can be effectively used as highly specific and robust tool and has recently been implemented in the diagnostic path of patients with rare disorders [19,20,21]. Sotos syndrome is the first overgrowth condition for which a specific DNAm signature was identified [22]. 

Genome-wide DNAm profiling is becoming a powerful functional tool in clinical setting [23] as a second-tier diagnostic strategy for the classification of VoUS, allowing to validate or rule out the clinical suspicion, and even confirm a diagnosis in exceptional cases, such as in individuals carrying hypomorphic or mosaic alleles [13]. This approach has successfully been applied to solve the clinical significance of a subset of *NSD1* VoUS [22]. Here, we employed microarray DNAm profiling to classify a variegated set of *NSD1* VoUS and private variants identified in the routine diagnostic setting. These cases included both subjects with features suggestive of Sotos syndrome who were assessed by targeted sequencing, and individuals with a non-suggestive phenotype who had been investigated by using an agnostic approach (exome sequencing). 

## 2. Methods

### 2.1. Study Cohort

All subjects were originally enrolled in the context of a routine diagnostic activity. Clinical data, pictures, DNA specimens and other biological material were collected, used, and stored after signed informed consents from the participating subjects/families were collected. Permission to publish clinical pictures was obtained for all subjects reported in Figure 1.

Six individuals (S1 to S6) presented with both a clinical diagnosis of Sotos syndrome and a molecularly diagnosed pathogenic variant in *NSD1*, either previously reported or considered *bona fide* as pathogenic. Their clinical and molecular characterization is summarized in Table 1. A second cohort of 11 individuals (S7 to S17) having a clinical suspicion of Sotos syndrome and an unclassified or a previously unreported variant in *NSD1* made up the testing group. Their clinical and molecular characterization is summarized in Table 1, while their detailed description is reported in the Appendix A. 

### 2.2. Molecular Genetics

In subjects S7 to S17, all variants were identified through massive parallel sequencing using different approaches, including target panel resequencing (multigene panel or clinical exome) or trio-based exome sequencing, after the exclusion of any clinically relevant structural variant by high-resolution CMA-array. The identified *NSD1* variants with their functional annotation are reported in Table 2. These variants and a general overview of the distribution of the *NSD1* variants annotated in ClinVar and gnomAD plotted taking into account the functional domain organization of the protein are shown in Figure 2. Their segregation, when available, has been confirmed by Sanger sequencing. 

### 2.3. Methylation Analysis

Peripheral blood DNA was extracted using standard techniques. Bisulphite conversion was performed, and samples were analyzed using Infinium Methylation EPIC BeadChip (Illumina, San Diego, CA, USA), according to the manufacturer’s protocol. The 17 samples with *NSD1* variants were analyzed, alongside 295 controls. 

IDAT files containing methylated and unmethylated signal intensities were imported into R V.4.2.1 for subsequent analysis using the *ChAMP* package (V.2.26.0) [24]. Firstly, intensity values were corrected for background using the Illumina’s method implemented in the minfi package [25] and evaluated for differences in cell type composition [26]. Afterwards, probes located on the X/Y chromosomes or known to cross-react with chromosomal locations other than their target regions, containing SNPs at or near the CpG sites, and having a detection *p*-value > 0.01 were excluded, resulting in 721,325 high-quality probes used in the subsequent analyses. Normalized methylation levels (β values) for each sample were compared using the established DNAm signature for Sotos syndrome (considering 110 high-quality probes out of 112) [17], and clustered by means of multidimensional scaling (MDS), considering the pair-wise Euclidean distances between samples. The training of the Support Vector Machine (SVM) machine learning (ML)-based classifier was performed with a linear kernel using the *e1071* R package V.1.7 and nu-classification option. To determine the best hyperparameter and measure the accuracy of the model, the whole dataset was split into a training set (75% of samples) and a test set (25% of samples), and a 5-fold cross-validation was performed during the training process. This procedure was repeated four times to verify that each sample was used at least once for testing. Finally, a SMOTE (*imbalance* R package V.1.0.2.1) oversampling technique was carried out to overcome class imbalance between affected and control individuals used in the training process [27]. Scores from the SVM classifier below 0.25 were considered as control samples, scores from 0.25 to 0.50 were considered inconclusive findings, while scores > 0.50 were deemed to behave as *NSD1* pathogenic variants.

## 3. Results

### 3.1. DNAm profiling allows the classification of rare/private NSD1 variants, confirming or ruling out the diagnosis of Sotos syndrome

Six individuals diagnosed with Sotos syndrome and carrying pathogenic *NSD1* mutations (Table 1) were initially tested to confirm the robustness and specificity of the previously identified disease-specific signature [17]. Subsequently, a MDS analysis considering the six previously confirmed Sotos samples, 11 affected individuals with overlapping clinical phenotypes suggestive of Sotos syndrome (Table 1, Figure 1, Appendix A) and found heterozygous for an unclassified or previously unreported *NSD1* variant, and 295 controls (including 213 neurotypical individuals and 82 affected subjects by different genetic diseases) was performed (Figure 3A). The analyses showed a clear clustering of seven subjects carrying missense/small in-frame deletion variants (S7, S8, S10-S14) together with the Sotos syndrome cohort. On the other hand, three individuals, including two subjects with de novo missense changes (S9 and S15) and one individual with a truncating variant inherited by an unaffected parent (S16), mapped far from the Sotos syndrome cluster and more closely to controls. S17, who carried a rare silent change (c.1782T > C), for which segregation was initially not available, clustered within the control group, ruling out pathogenicity.

To functionally classify the *NSD1* VoUS, an SVM-based ML classifier was applied, taking advantage of the established Sotos syndrome-specific episignature. By training a SVM model on five *bona fide* pathogenic variants versus our internal control database (~300 samples), the remaining sample with a likely pathogenic variant (S3) was validated, and the 11 tested samples were unambiguously classified (Table 2; Figure 3B). Pathogenic scores (>0.95) were obtained in the seven samples that were identified to cluster together with the Sotos syndrome cohort by MDS analysis. A pathogenic score (0.84) was also obtained for S9, who was definitely reclassified as affected with Sotos syndrome. Again, a control score (<0.10) was reached for S17. The two remaining individuals (S15 and S16) having an ambiguous DNAm profile at the MDS plot were definitively reclassified using the ML-classifier, which ruled out a diagnosis of Sotos syndrome, assigning benign probability scores (<0.25) to their variants.

### 3.2. Clinical and Molecular Re-Evaluation of Subjects Carrying Reclassified NSD1 Variants

We revised the clinical and molecular characteristics of all the individuals carrying a reclassified *NSD1* variant. All subjects with a reclassified pathogenic variant showed a clinical phenotype fulfilling all the elements of the triad requested to suspect a clinical diagnosis of Sotos syndrome, including overgrowth (8/8) and macrocephaly (6/7), suggestive facies (8/8), and DD/ID (8/8). In particular, the cardinal facial features characterized by broad and prominent forehead (8/8), dolichocephaly (7/8), downslanted palpebral fissures (8/8), long and narrow face (8/8), and long/pointed chin (8/8) were invariably observed. On the other hand, the three individuals carrying variants reclassified as non-pathogenic for Sotos syndrome showed a phenotype that only partially fit a clinical diagnosis of the disorder, having less marked suggestive facial features, and not showing hypotonia; among these, S15 and S16 also did not show ID (Table 1). Two of the three subjects (S15 and S17) presented with overgrowth, while the co-occurrence of cystic fibrosis in the third individual (S16) (see Appendix A) could justify his growth within the normal range.

*NSD1* is characterized by an overall PLP clustering score of 0.75, and regions between residues 1,549-1,624 and 1,633-2,241 are identified as significant regions with enriched PLP variants. As shown in Figure 2, all the tested VoUS reclassified as pathogenic by DNAm profiling were predicted as pathogenic by MutScore (>0.73) and map in regions that are classified as significant PLP clusters [9], corresponding to the PHD, PWWP and SET functional domains. Specifically, the majority were located within the PHD and SET domains, mainly at the highly conserved cysteine residues (Figure 2). Among the other missense changes, the variant predicting the Phe-to-Ser change at codon 2,043 within the SET domain (S11) also shows a clear match with the disease-specific DNAm signature, highlighting an important role of this highly conserved residue for the proper stability/function of this domain. Similarly, p.Trp1806Cys (S7) involves a key residue within one of the two PWWP domains (PWWP2). Consistently, subsequent segregation analysis performed in families of S8, S9, S10, S12, S13, and S14 confirmed the de novo origin of the identified variants, further supporting their pathogenic role. When considering the variants that did not fit the DNAm signature, the c.7393C > T (p.Gln2465*) (S16) was observed to map within a functionally uncharacterized region at the C-terminus. Of note, this variant was inherited from an apparently normal father, who only shows macrocrania. Familial inheritance of Sotos syndrome has rarely been observed. Tatton-Brown et al. (2005) [4] screened >300 parents of *NSD1* mutation-positive individuals, identifying a mutation in only 11, all of whom had previously been clinically diagnosed with familial Sotos syndrome, and indicating that incomplete penetrance is an extremely rare event for pathogenic *NSD1* variants. S17 presented with suggestive features of Sotos syndrome and macrosomia with macrocrania, and had a silent change in *NSD1*. The lack of signature in this patient allowed us to exclude a role of the observed variant in the pathogenesis of his condition. In line with our conclusions, subsequent segregation analysis documented that the variant was transmitted from a mother with isolated macrocrania and tall stature. Finally, the application of DNAm profiling in S15 allowed to reclassify the de novo c.1096G > A substitution (p.Val366Met), which affects the PWWP1 domain, rarely occurs in the general population (rs1172667661), and has an uncertain interpretation by ClinVar (VCV000451652). Of note, conflicting annotation was also reported by MutScore, which assigned a high score for pathogenicity, though it does not map within a region with a significant PLP clustering (Figure 2).

### 3.3. Robustness of the SVM Classifier and Sample Size Requirement

To assess how the training cohort size affects the sensitivity and prediction confidence (i.e., probability score) of the classifier, the ML-based classifier training was performed with subsets of the original training cohort. Using different sizes (3, 4 or 5 randomly selected samples with pathogenic *NSD1* variants), we performed different rounds of training, testing on the same time if prediction confidence could be further improved using a more balanced dataset by means of oversampling (see Methods). The sensitivity and prediction confidence of different classifiers were compared, showing no substantial differences among probability scores obtained from the models using the full cohort or a minimal cohort of three samples with pathogenic variants as training sets, with class imbalance correction (Table 3). On the other hand, the analysis documented that the performance of the ML-based classifier was relevantly affected by the oversampling procedure, resulting crucial for the classification of three testing individuals (S9, S13, and S15) using all the different training cohorts (Table 3). This is in line with the higher accuracy of the model that was observed following oversampling (0.97 vs. 0.92, with and without oversampling, respectively).

## 4. Discussion

DNAm profiling has emerged as a highly informative tool for “functional” validation of putative disease-causing variants. The present findings further confirm the diagnostic utility of this technology in clinical routine diagnostics in solving “real” cases of VoUS for which “traditional” validation using in silico, in vitro and in vivo strategies or segregation analysis are not practicable. The use of DNAm profiling in this case series allowed us to definitively confirm pathogenicity of a new subset of missense and small in-frame deletions that were classified as VoUS and in a small group of previously unreported variants, without recurring to evidence-based of multiple observations eventually classifying them in ClinVar. 

The routine use of massive parallel sequencing techniques (targeted sequencing, exome/genome sequencing) has revealed an emerging problem on VoUS interpretation in either highly suggestive or nonsuggestive associated phenotypes. DNAm profiling in a subset of conditions for which a signature has been established is becoming a reflex assay for the assessment of functional significance and reclassification of VoUS. This approach has the advantage of allowing a direct assessment on the primary sample, not requiring a new sampling or a different tissue analysis. If widely applied, this strategy could overcome the “traditional” validation approaches through functional assessment in various model systems, or clinical confirmation in multiple unrelated patients/families, which are usually unavailable for routine screening due to the rarity of these conditions. In the present series, five individuals (S7, S8, S12–S14) carried previously unreported variants, while three subjects (S9-S11) were heterozygous for variants reported as VoUS or having conflicting interpretation in ClinVar. All of them were successfully reclassified as pathogenic by applying this strategy. 

Our classifier also managed to exclude pathogenicity of the remaining three assessed cases in the context of Sotos syndrome. One truncating *NSD1* variant (S16) was inherited from an apparently unaffected father. S16 showed a co-occurring pathogenic homozygous variant in *CFTR* (p.Phe508del). She clinically presented with cystic fibrosis and relative macrocrania, which was also observed in her transmitting father. DNAm profiling allowed us to exclude a dual molecular diagnosis expressed with a blended phenotype. The application of the classifier also allowed to directly verify pathogenicity in a variant for which segregation was not available (S11) and exclude pathogenicity in a subject with a highly suggestive presentation (S17). In the latter, thanks to the negative finding, this individual is likely not to have either a deep intronic or an *NSD1* intragenic deletion, avoiding further targeted analyses directed to explore other genomic events involving *NSD1*. Subsequent segregation analysis revealed variant transmission from the mother, who presented with macrocrania, tall stature, and mild learning difficulties, during the scholar period. Finally, S15 presented with many typical facial features of Sotos syndrome (macrocrania, long and triangular face, downslanted palpebral fissure, and deep-set eyes) but lacked two main features (dolicocephaly and pointed chin). Moreover, she did not show hypotonia and ID, thus lacking a main sign to clinically diagnose Sotos syndrome. Her variant was also reported in both ClinVar with uncertain significance and in general population as a rare allele. This variant showed a MutScore value greater than the suggested pathogenicity threshold (0.73) [9]. We cannot exclude a hypomorphic effect for the variants identified in S15–S17 on *NSD1* function that might underlie or contribute to the isolated macrocephaly observed in the three patients and their transmitting parents. Our findings, however, definitively exclude a pathogenic role of these variants in Sotos syndrome pathogenesis.

Of note, the present analyses showed that the use of even a small number (down to three) of samples with *bona fide* pathogenic variants can successfully be used to effectively train a ML-based classifier based on the existing Sotos syndrome-specific DNAm signature, allowing VoUS classification. 

In conclusion, the available episignature for Sotos syndrome has demonstrated to be a robust and reliable tool to quickly rule out or confirm a clinical suspicion of this disorder. Episignatures have the great advantage to overcome many technical limitations observed in current sequencing techniques (e.g., the lack of coverage of introns in the search of deep intronic pathogenic variants, promoters, variants involving regulatory regions). If properly used, this assay might be considered as a first-tier analysis in subjects with specific clinical suspicion in whom a condition caused by a gene involved in the epigenetic machinery is suspected. As exemplified in our case (S17), even in the presence of a highly suggestive Sotos syndrome phenotype, this assay could easily rule out the clinical diagnosis without proceeding with more complex targeted analyses directed to explore other genomic events involving *NSD1*. The application of DNAm analysis in diagnostics, VoUS classification, exploration of disease pathophysiology, as well as its emerging use in biomarker identification applied to precision medicine are becoming progressively more promising and are expected to be routinely applied in the next future.

## Figures and Tables

**Figure 1 genes-13-02163-f001:**
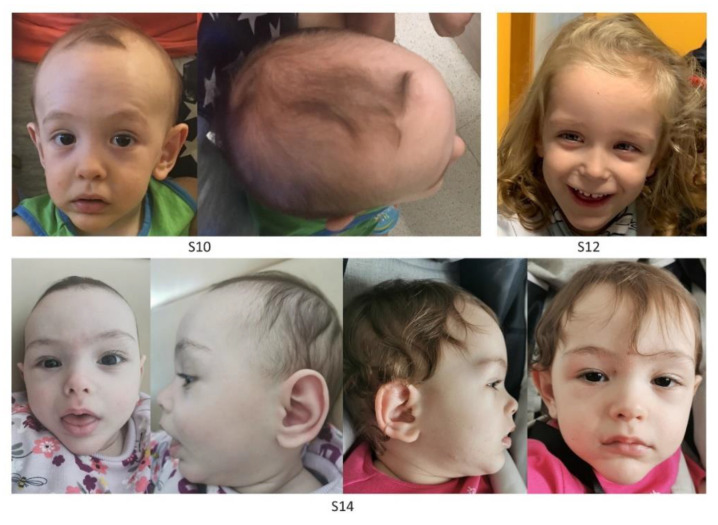
**Facies of three subjects with clinical diagnosis of Sotos syndrome carrying functionally unclassified *NSD1* variants.** Clinical features of S10 (1 year), S12 (2 years) and S14 (6 months, left panels; 1 year and 3 months, right panels) showing typical Sotos syndrome characteristics. All subjects show macrocephaly, dolichocephaly, broad and prominent forehead, sparse fronto-temporal hair, downslanted palpebral fissures, long and narrow face, and a long/pointed chin.

**Figure 2 genes-13-02163-f002:**
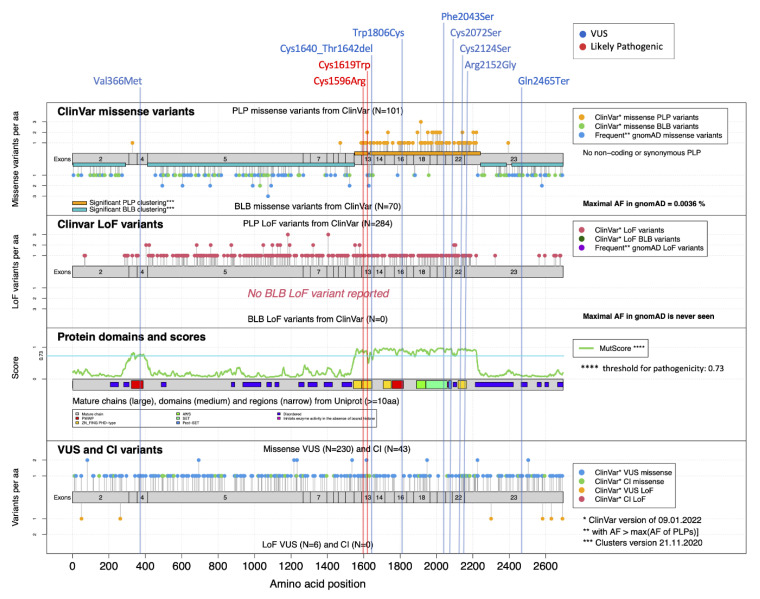
**Mutational landscape for *NSD1***. MutLand plot for *NSD1* (NM_022455.4) [9], showing exons, functional domains and variants reported in ClinVar. Location of the variants included in the present study are also shown, as per different ACMG classification (blue, VoUS; red, likely pathogenic). Positions of the missense and LoF *NSD1* variants reported in ClinVar known to be either pathogenic/likely pathogenic (PLP) or benign/likely benign (BLB) are shown (top diagrams), whereas VoUS or variants having conflicting interpretation (CI) reported by ClinVar, either missense or LoF, are shown in the bottom diagram. The cartoons highlight the regions for which MutScore detects significant PLP clustering (orange horizontal bars). A diagram reporting the functional domains of the protein (PHD, yellow; PWWP, red; SET, light green), and plotting the MutScore referred for the entire protein sequence is also shown (third plot from the top).

**Figure 3 genes-13-02163-f003:**
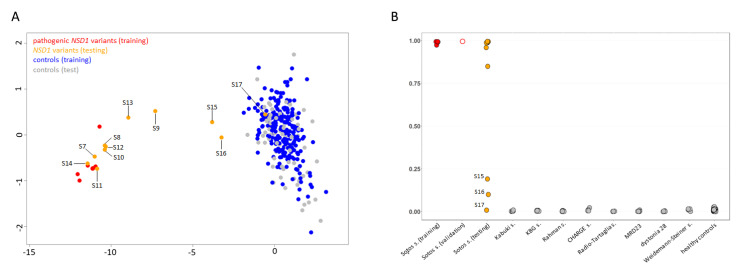
**DNAm array analyses.** (**A**) DNAm profiles in individuals carrying *NSD1* variants versus a control cohort. Taking advantage of DNAm levels from the established episignature, MDS plot was used to test/validate sample clustering with respect to the Sotos syndrome training cohort (red, solid circles) and a control group including 295 healthy individuals and subjects with rare neurodevelopmental disorders (blue, solid circles). Tested VoUS are depicted in orange (solid circles), while control subset used for testing are in grey (solid circles). (**B**) Plot for SVM probability scores from the developed ML-based classifier, showing classification results for the tested samples (VoUS, n = 8, orange solid circles; validation samples, n = 1, red open circles), and the in-house control cohort (grey = testing).

**Table 1 genes-13-02163-t001:** **Clinical and molecular features of the subjects included in the study**.

ID (Age at Last Examination, Years)		S1 (2.9)		S2 (23)	S3 (3)		S4 (2)	S5 (1)	S6 (2)	Tot
Pathogenic *NSD1* Variant		Deletion of Exons 14 and 15		c.5147-2A > G	c.4638T > A (p.Cys1546*)		c.6010-2A > G	c.6070C > T (p.Gln2024*)	c.2370delA (p.Lys791fs*15)	
Growth	Birth weight (cent)	95°		>97°	>97°		50°	14°	>97°	3/6
	Birth length (cent)	>97°		>97°	97°		90°	10°	>97°	4/6
	Birth OFC (cent)	>97°		>97°	>>97°		50-75°	64°	>97°	4/6
	Postnatal weight (cent)	>97°		>97°	97°		>97°	>97°	75°-90°	5/6
	Postnatal height (cent)	>97°		75°	97°		90-97°	75-97°	>97°	4/6
	Postnatal OFC (cent)	>97°		>97°	>>97°		>97°	>97°	75°-90°	5/6
Development	DD/ID	moderate		moderate	mild		moderate	mild	moderate	6/6
Neurology	Hypotonia (Ho)	yes		yes	yes		yes	yes	yes	6/6
	Seizures/EEG anomalies	no		no	no		no	yes, febbrile	no	1/6
	Brain anomalies	no		mild ectasia of trigone of lateral ventricles; arachnoid cyst	reduced white matter thickness and hyperintensity in the subcortical white matter; wide aspect of the periencephalic spaces; squared aspect of the frontal horns; thin corpus callosum; multicystic aspect of the pineal gland		right posterior fronto-opercular polymicrogiria; thin corpus callosum; thin optic nerves	NA	no	3/5
Craniofacial	Dolichocephaly	yes		yes	yes		yes	yes	yes	6/6
	Prominent forehead	yes		yes	yes		yes	yes	yes	6/6
	Long/triangular face	yes		yes	yes		yes	yes	yes	6/6
	Downslanted palpebral fissures	yes		yes	yes		yes	yes	yes	6/6
	Deep set eyes	no		yes	yes		yes	yes	yes	5/6
	Depressed nasal bridge	no		yes	yes		yes	yes	yes	5/6
	Everted lips	yes		yes	yes		yes	yes	yes	6/6
	Pointed/prominent chin	yes		yes	yes		yes	yes	yes	6/6
Musculo-skeletal	Advanced bone age		yes		yes	yes	NA	NA	NA	3/3
	Long hands	yes		yes	no		yes	yes	yes	5/6
	Joint laxity	yes		yes	no		yes	yes	yes	5/6
Cardiac		ASD OS		no	mild aortic insufficiency		no	transient neonatal bradycardia; ASD OS	moderate tricuspid insufficiency; ASD OS	4/6
Other		no		club feet scoliosis	no		unilateral cryptorchidism; hypoglycemia at birth	prolonged neonatal jaundice	OSAS, mild hepatomegaly	
**ID** **(Age at Last Examination, Years)**		**S7** **(11)**	**S8** **(1)**	**S9** **(2)**	**S10** **(1)**	**S11** **(0.9)**	**S12** **(2)**	**S13** **(4)**	**S14** **(0.6)**	**Tot**
**Identified *NSD1* Variant**		**c.5418G > T** **(p.Trp1806Cys)**	**c.4857T > G** **(p.Cys1619Trp)**	**c.6454C > G** **(p.Arg2152Gly)**	**c.4786T > C** **(p.Cys1596Arg)**	**c.6128T > C** **(p.Phe2043Ser)**	**c.4917_4925delCTGTATAAC (p.Cys1640_Thr1642del)**	**c.6371G > C (p.Cys2124 Ser)**	**c.6215G > C (p.Cys2072Ser)**	
Growth	Birth weight (cent)	25°	25°	50°	90°	86°	97°	>97°cc	66°	2/8
	Birth length (cent)	25°	50°	75°	97°	76°	NA	97°	71°	2/7
	Birth OFC (cent)	25°	75°	75°	90°	90°	>97°	NA	75°	1/7
	Postnatal weight (cent)	75°	>97°	97°	>97°	>97°	>97°	90°-97°	71°	6/8
	Postnatal height(cent)	97°	>97°	>97°	>97°	>97°	>97°	>97°	99°	8/8
	Postnatal OFC (cent)	>97°	97°	>97°	NA	97°	>97°	>97°	88°	6/7
Development	DD/ID	borderline	mild	mild	moderate	moderate	mild	mild	mild	8/8
Neurology	Hypotonia	yes	no	yes	yes	yes	yes	yes	yes	7/8
	Seizures/EEG anomalies	no	no	no	no	yes	no	no	no	1/8
	Brain anomalies ^#^	yes	no	no	yes	yes	yes	NA	yes	5/7
Craniofacial	Dolichocephaly	yes	yes	yes	yes	yes	no	yes	yes	7/8
	Prominent forehead	yes	yes	yes	yes	yes	yes	yes	yes	8/8
	Long/triangular face	yes	yes	yes	yes	yes	yes	yes	yes	8/8
	Downslanted palpebral fissures	yes	yes	yes	yes	yes	yes	yes	yes	8/8
	Deep set eyes	yes	yes	yes	no	yes	yes	yes	yes	7/8
	Depressed nasal bridge	no	no	no	yes	no	no	yes	yes	3/8
	Everted lips	no	no	no	yes	yes	no	no	yes	3/8
	Pointed/prominent chin	yes	yes	yes	yes	yes	yes	yes	yes	8/8
Musculo-skeletal	Advanced bone age	yes	yes	yes	NA	yes	yes	yes	NA	6/6
	Long hands	yes	yes	yes	no	yes	no	no	yes	5/8
	Joint laxity	yes	no	no	yes	yes	yes	yes	yes	6/8
Cardiac		no	VSD (muscular); DPV	no	no	ASD	no	mitral valve prolapse		3/8
Other		scoliosis		born after ART	scoliosis	feet oedema; cutis laxa; GER; scoliosis	recurrent otitis	mild renal asymmetry	scoliosis	
**ID** **(Age at Last Examination, Years)**				**S15** **(7)**	**S16** **(0.8)**			**S17** **(18.7)**	**Tot**	
**Identified *NSD1* Variant**				**c.1096G > A** **(p.Val366Met)**	**c.7393C > T** **(p.Gln2465*)**			**c.1782T > C** **(p.Pro594=)**		
Growth	Birth weight (cent)			75°	63°			>97°	1/3	
	Birth length (cent)			75°	NA			>97°	1/2	
	Birth OFC (cent)			97°	54°			>97°	2/3	
	Postnatal weight (cent)			75°	67°			>97°	1/3	
	Postnatal height(cent)			75°	16°			>97°	1/3	
	Postnatal OFC cent)			>97°	relative macrocrania			>97°	2/3	
Development	DD/ID			no	NA			yes	1/2	
Neurology	Hypotonia (Ho)			no	no			yes	1/3	
	Seizures/EEG anomalies			yes ^	no			yes	2/3	
	Brain anomalies			no	no			no	0	
Craniofacial Craniofacial	Dolichocephaly			no	no			yes	1/3	
	Prominent forehead			yes	yes			yes	2/3	
	Long/triangular face			yes	no			yes	2/3	
	Downslanted palpebral fissures			no	no			yes	1/3	
	Deep set eyes			yes	no			no	1/3	
	Depressed nasal bridge			yes	yes			no	2/3	
	Everted lips			yes	no			yes	2/3	
	Pointed/prominent chin			no	no			yes	1/3	
Musculo-skeletal	Advanced bone age			NA	NA			yes	1	
	Long hands			no	no			yes	1/3	
	Joint laxity			no	no			no	0	
Cardiac				no	no			no	0	
other				fetal pads	father with macrocephaly; homozygosity for F508; short limbs; dup. Xq21.1 inherited from the mother			scoliosis		

^ Generalized, tonic-clonic seizures, without fever, with EEG abnormalities. ^#^ Detailed description in supplemental clinical reports. ASD OS, atrial septal defect ostium secundum type; DD, developmental delay; DPV, dysplastic pulmonary valve; ID, intellectual disability; OSAS, obstructive sleep apnea syndrome; VSD, ventricular septal defect; GER, gastroesophageal reflux; ART, assisted reproductive technique; OFC, occipito-frontal circumference; NA, not assessed.

**Table 2 genes-13-02163-t002:** **Functional annotation of the *NSD1* variants included in the study**.

Patient ID	*NSD1* Variant (NM_022455.5)	Type of Variant ^a^	Exon	Inheritance	rsID (maxAF)	ACMG Classification ^b^	Domain ^c^	CADD Score (PHRED)	Sample Set	SVM Score
S1	5q35.3 microdeletion	SV—multi-exon deletion	14-15	NA	NR	5—P	-	-	training	0.99
S2	c.5147-2A > G	SNV—splice site	15	de novo	NR	4—LP	-	35.0	training	0.98
S3	c.4638T > A, p.(Cys1546 *)	SNV—stop gain	11	de novo	NR	4—LP	-	35.0	validation	0.99
S4	c.6010-2A > G	SNV—splice site	20	de novo	NR	4—LP	-	34.0	training	0.97
S5	c.6070C > T; p.(Gln2024 *)	SNV—stop gain	20	de novo	NR	5—P	-	43.0	training	0.99
S6	c.2370delA; p.(Lys791Serfs*16)	SNV—frameshift	5	NA	NR	4—LP	-	32.0	training	0.99
S7	c.5418G > T; p.(Trp1806Cys)	SNV—nonsynonymous	16	de novo ^d^	NR	3—VoUS	PWWP2	32.0	testing	0.99
S8	c.4857T > G; p.(Cys1619Trp)	SNV—nonsynonymous	13	de novo ^d^	NR	4—LP	PHD-type1	26.5	testing	0.98
S9	c.6454C > G; p.(Arg2152Gly)	SNV—nonsynonymous	22	de novo ^d^	NR	3—VoUS	PHD-type4	24.9	testing	0.84
S10	c.4786T > C; p.(Cys1596Arg)	SNV—nonsynonymous	13	de novo ^d^	NR	4—LP	PHD-type1	28.6	testing	0.98
S11	c.6128T > C; p.(Phe2043Ser)	SNV—nonsynonymous	20	NA	NR	3—VoUS	SET	30	testing	0.99
S12	c.4917_4925delCTGTATAAC; p.(Cys1640_Thr1642del)	in-frame deletion	13	de novo ^d^	NR	3—VoUS	PHD-type2	22.3	testing	0.98
S13	c.6371G > C; p.(Cys2124 Ser)	SNV—nonsynonymous	22	de novo ^d^	rs757818289 (no frequency)	3—VoUS	PHD-type4	27.7	testing	0.95
S14	c.6215G > C p.(Cys2072Ser)	SNV—nonsynonymous	21	de novo ^d^	NR	3—VoUS	Post-SET	28.5	testing	0.99
S15	c.1096G > A; p.(Val366Met)	SNV—nonsynonymous	4	de novo	rs1172667661 (0.000003976)	3—VoUS	PWWP1	26.0	testing	0.19
S16	c.7393C > T; p.(Gln2465*)	SNV—stop gain	23	paternal	NR	3—VoUS	-	39.0	testing	0.09
S17	c.1782T > C; p.(Pro594 = )	SNV—synonymous	5	maternal	rs200002555 (0.00001171)	1—B	-	7.5	testing	0.08

^a^ SNV, single nucleotide variant; SV, structural variant. ^b^ B, benign; LP, likely pathogenic; P, pathogenic; VoUS, variant of unknown significance. ^c^ PHD, plant homeodomain; PWWP1, Pro-Trp-Trp-Pro domain; SET, Su(var)3–9, Enhancer-of-zeste, Trithorax domain. ^d^ Segregation analysis performed *a posteriori*. NA, not available; NR, not reported.

**Table 3 genes-13-02163-t003:** **Training sample size requirement for *NSD1* variant classification using the SVM classifier**.

Sample ID	SMOTE-Balanced Dataset			Unbalanced Dataset		
	Dataset 1	Dataset 2	Dataset 3	Dataset 1	Dataset 2	Dataset 3
S7	P (0.982)	P (0.987)	P (0.994)	P (0.812)	P (0.842)	P (0.870)
S8	P (0.971)	P (0.979)	P (0.982)	P (0.727)	P (0.773)	P (0.778)
S9	P (0.830)	P (0.840)	P (0.849)	I (0.262)	I (0.306)	I (0.311)
S10	P (0.977)	P (0.980)	P (0.989)	P (0.748)	P (0.783)	P (0.788)
S11	P (0.989)	P (0.991)	P (0.993)	P (0.791)	P (0.813)	P (0.820)
S12	P (0.975)	P (0.991)	P (0.988)	P (0.716)	P (0.766)	P (0.768)
S13	P (0.952)	P (0.968)	P (0.969)	I (0.482)	P (0.550)	P (0.557)
S14	P (0.974)	P (0.992)	P (0.997)	P (0.838)	P (0.870)	P (0.870)
S15	C (0.172)	C (0.179)	C (0.192)	I (0.473)	P (0.522)	P (0.522)
S16	C (0.119)	C (0.106)	C (0.100)	C (0.029)	C (0.031)	C (0.033)
S17	C (0.019)	C (0.018)	C (0.007)	C (0.006)	C (0.007)	C (0.007)
S3	P (0.986)	P (0.990)	P (0.993)	P (0.809)	P (0.843)	P (0.845)

P, pathogenic scores (>0.50); I, inconclusive scores (>0.25, <0.50); C, control scores (<0.25). Datasets 1, 2 and 3 include 3, 4 and 5 randomly selected samples (*bona fide* pathogenic variants) used as training set, respectively. Each dataset was assessed either by using oversampling (SMOTE algorithm) or without balancing the different classes (training/testing). Samples S3 (*bona fide* pathogenic *NSD1* variant) was used as validation samples in each testing round.

## Data Availability

The data that support the findings of this study are available on request from the corresponding author (M.T.). The data are not publicly available due to due to privacy/ethical restrictions.

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
