# Peer review of "Genome-Wide DNA Methylation Profiling Solves Uncertainty in Classifying NSD1 Variants"

_genes, 2022, doi:10.3390/genes13112163_

Round 1

Reviewer 1 Report

In this study, the authors evaluated the pathogenicity of the variants by microarray DNAm profiling in a group of patients with Sotos syndrome carrying VUS or previously unreported variants in the NSD1 gene. This study offers an innovative perspective to bring new insights in this subject. It may lead to new studies to evaluate VUS variants and a genotype-phenotype correlation.

The authors must review the article again in terms of grammatical errors.

There are some grammatical errors;

-A large proportion of overgrowth syndromes is (are) caused by pathogenic variants in genes (line 91)

-Sotos syndrome is the first overgrowth condition 105 for which a specific DNAm signature was identified (line 105).

-These cases included either subjects with features suggestive of Sotos syndrome who were assessed by targeted sequencing, and individuals with a non-suggestive phenotype who had been investigated by using an agnostic approach (exome sequencing). (line 113-115)

-In subjects S7 to S17, all variants were identified through massive parallel sequencing using different approaches (such?) as target panel resequencing…..

-To determine the best hyperparameter and measure the accuracy of the model, the whole dataset was split in (into???) a training set….

-confirming or ruling out  (the???) diagnosis of Sotos syndrome.

-Pathogenic scores (>0.95) were obtained in all the seven samples previously clustering in the Sotos syndrome cohort by MDS analysis. (check it)

- which ruled rule (x2?????) out a diagnosis of Sotos syndrome, assigning pathogenicity scores <0.25 to their variants. (check it)

- This is in line with the higher accuracy of the model that was observed following oversampling (0.97 vs 0.92).(check it)

- DNAm profiling in a subset of conditions for which a signature has been established is becoming a reflex assay for the assessment of functional significance and reclassification of VoUS and has the advantage of allowing a direct assessment on the primary sample, not requiring a new sampling or a different tissue analysis. (a complex sentence)

- Our classifier also allowed to directly verify pathogenicity in a variant for whom segregation was not available (S11) and exclude possible splicing effect in a subject with a highly suggestive presentation (S17).(check it)

- Subsequent segregation analysis revealed a maternal transmission of the variant, who presented with macrocrania and tall stature and mild learning difficulties during the scholar period.(check it)

- (macrocrania, long and triangular face, downslanted palpebral fissure, and deep set yes (eyes??))

- Although we cannot exclude a hypomorphic effect for the variants identified in S15-S17 on NSD1 function that might underlie isolated macrocephaly observed in the three patients and their transmitting parents, we definitively excluded their major role in Sotos syndrome pathogenesis. (a complex sentence)

-Of note, our analyses also showed that a small number (down to three) of samples with bona fide pathogenic variants could be sufficient to effectively train a ML-based classifier based on the existing Sotos syndrome-specific DNAm signature, allowing us to classify a wide range of VoUS associated to atypical clinical presentations, or lacking familiar information. (check it)

-In conclusion, the available episignature for Sotos syndrome has demonstrated to be a robust and reliable tool to quickly rule out or confirm…. (check it)

-Its role in diagnostics, VoUS classification, exploration of disease pathophysiology, and the emerging use in the search of new treatment strategies is now becoming more and more promising and will take a major role in the next future. (check it)

Author Response

We thank the reviewer for her/his positive feedback.

Remarks

There are some grammatical errors:

(1) “A large proportion of overgrowth syndromes is (are) caused by pathogenic variants in genes (line 91)

(2) “Sotos syndrome is the first overgrowth condition 105 for which a specific DNAm signature was identified (line 105)

(3) “These cases included either subjects with features suggestive of Sotos syndrome who were assessed by targeted sequencing, and individuals with a non-suggestive phenotype who had been investigated by using an agnostic approach (exome sequencing). (line 113-115)

(4) “In subjects S7 to S17, all variants were identified through massive parallel sequencing using different approaches (such?) as target panel resequencing…..”

(5) “To determine the best hyperparameter and measure the accuracy of the model, the whole dataset was split in (into???) a training set….”

(6) “confirming or ruling out (the???) diagnosis of Sotos syndrome.”

(7) “Pathogenic scores (>0.95) were obtained in all the seven samples previously clustering in the Sotos syndrome cohort by MDS analysis. (check it)

(8) “which ruled rule (x2?????) out a diagnosis of Sotos syndrome, assigning pathogenicity scores <0.25 to their variants. (check it)

(9) “This is in line with the higher accuracy of the model that was observed following oversampling (0.97 vs 0.92).(check it)

(10) “DNAm profiling in a subset of conditions for which a signature has been established is becoming a reflex assay for the assessment of functional significance and reclassification of VoUS and has the advantage of allowing a direct assessment on the primary sample, not requiring a new sampling or a different tissue analysis. (a complex sentence)

(11) “Our classifier also allowed to directly verify pathogenicity in a variant for whom segregation was not available (S11) and exclude possible splicing effect in a subject with a highly suggestive presentation (S17).(check it)

(12) “Subsequent segregation analysis revealed a maternal transmission of the variant, who presented with macrocrania and tall stature and mild learning difficulties during the scholar period.(check it)

(13) “(macrocrania, long and triangular face, downslanted palpebral fissure, and deep set yes (eyes??))

(14) “Although we cannot exclude a hypomorphic effect for the variants identified in S15-S17 on NSD1 function that might underlie isolated macrocephaly observed in the three patients and their transmitting parents, we definitively excluded their major role in Sotos syndrome pathogenesis. (a complex sentence)

(15) “Of note, our analyses also showed that a small number (down to three) of samples with bona fide pathogenic variants could be sufficient to effectively train a ML-based classifier based on the existing Sotos syndrome-specific DNAm signature, allowing us to classify a wide range of VoUS associated to atypical clinical presentations, or lacking familiar information. (check it)

(16) “In conclusion, the available episignature for Sotos syndrome has demonstrated to be a robust and reliable tool to quickly rule out or confirm…. (check it)

(17) “Its role in diagnostics, VoUS classification, exploration of disease pathophysiology, and the emerging use in the search of new treatment strategies is now becoming more and more promising and will take a major role in the next future. (check it)

Authors’ reply: We thank the reviewer for her/his detailed assessment and for highlighting the above listed typos, misleading terminology and unclearly formulated sentences, which were amended in the revised version of the manuscript. As regarding the remark (1), we retained “is” in the sentence, as the verb refers to the third-person singular term “large proportion”.

(18) Are the methods adequately described? Must be improved

Authors’ reply: We revised the Methods section to improve its clarity. To satisfy this request, we re-wrote part of the methodology trying to simplifying it. It should be noted, however, that bioinformatics analyses cannot be properly described without using technical terminology. So we apologize with the reviewer but some technical terms had to be retained.

Reviewer 2 Report

The microarray DNAm profiling method is very useful for clinical diagnosis, but the stability and repeatability of the microarray technology should be validated in clinical laboratory.

Author Response

We thank the reviewer for the positive evaluation of this work.

Remarks:

(1) “The microarray DNAm profiling method is very useful for clinical diagnosis, but the stability and repeatability of the microarray technology should be validated in clinical laboratory.

Authors’ reply: We absolutely agree with the reviewer, and we truly think that the use of this tool in the diagnostic setting must follow a robust validation. Indeed, we are aware of guidelines and validation efforts devoted to quickly transfer this methodology in the clinical diagnostic practice (see: Sadikovic et al. 2021, Genet Med 23:1065-1074). At the same time, we are truly convinced that our work will be useful for empowering knowledge of this new tool to a wider audience of clinicians and geneticists.
